# Using wearable activity trackers for research in the global south: Lessons learned from adolescent psychotherapy research in Kenya

Natalie E. Johnson[1,2,3] [iD], Katherine E. Venturo-Conerly[1,4] and Thomas Rusch[5]

[1]Department of Research and Evidence, Shamiri Institute, Nairobi, Kenya; [2]Division of Clinical Epidemiology, Department of Clinical Research, University Hospital Basel, Basel, Switzerland; [3]Faculty of Medicine, University of Basel, Basel, Switzerland; [4]Department of Psychology, Harvard University, Cambridge, MA, USA and [5]Competence Center for Empirical Research Methods, WU Vienna University of Economics and Business, Vienna, Austria

## Perspective

**Keywords:**
mobile health technology; fitness trackers; sub-Saharan Africa; mental health; wearable

**Corresponding author:**
Natalie E. Johnson;
Email: natalieelaine.johnson@usb.ch

## Abstract

Wearable activity trackers have emerged as valuable tools for health research, providing high-resolution data on measures such as physical activity. While most research on these devices has been conducted in high-income countries, there is growing interest in their use in the global south. This perspective discusses the challenges faced and strategies employed when using wearable activity trackers to test the effects of a school-based intervention for depression and anxiety among Kenyan youth. Lessons learned include the importance of validating data output, establishing an internal procedure for international procurement, providing on-site support for participants, designating a full-time team member for wearable activity tracker operation, and issuing a paper-based information sheet to participants. The insights shared in this perspective serve as guidance for researchers undertaking studies with wearables in similar settings, contributing to the evidence base for mental health interventions targeting youth in the global south. Despite the challenges to set up, deploy and extract data from wearable activity trackers, we believe that wearables are a relatively economical approach to provide insight into the daily lives of research participants, and recommend their use to other researchers.

## Impact statement

There is a significant need for ecologically valid methods to measure the effects of mental health interventions on youth in the global south, where access to psychological care is limited. Wearables have emerged as powerful tools, providing high-resolution data on mental health markers. This perspective discusses the use of wearable activity trackers, or wearables, in the context of psychotherapy research with youth in Kenya. It highlights several key lessons learned and best practices that are relevant for global mental health researchers. We outline the challenges faced when using wearables in Kenya, such as device selection, procurement, technical issues, and participant support. It offers recommendations, including the importance of validating devices, establishing an efficient procurement process, providing onsite support, designating a full-time team member, and distributing standardized information sheets to participants. The impact of this article extends beyond Kenya, as it provides practical insights that can be applied to similar contexts globally. It underscores the importance of accurate and ecologically valid data collection in mental health research and encourages researchers to consider the benefits of wearable activity trackers while addressing the associated challenges. Ultimately, the lessons shared here can help researchers successfully implement global mental health studies with wearables, contributing to more robust research in these settings.

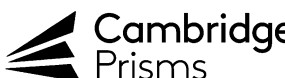

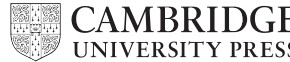

## Introduction

Approximately 90% of youth live in the global south, with children under 15 accounting for 41% of the population in sub-Saharan Africa (United Nations, Department of Economic and Social Affairs, Population Division, 2015). For these youth, access to psychological care is limited (Galagali and Brooks, 2020), although an estimated 10–15% of disease in these settings is neuropsychiatric (Patel, 2007). Thus, it is important to find economical and ecologically valid methods to measure the effects of mental health interventions. Wearable activity trackers (WATs) have emerged as powerful tools for health research in recent years, enabling the collection of high-resolution data on physical activity, sleep, and other health markers and behaviors (Ryan et al., 2019; Cho et al., 2020; Lewis et al., 2020; Semaan et al., 2020; Zahrt et al., 2023). While most research using these devices with youth has been conducted in high-income countries (HICs; Casado-Robles et al., 2022), there is

growing interest in using them to study health in the global south (Barteit et al., 2021; Neale et al., 2023).

When using these devices in HICs, researchers have met challenges with participant compliance, technology (e.g., synchronization), and logistical issues such as limited battery life (Harrison et al., 2014). One study identified the need for a patient-facing technology support desk when using WATs (Smuck et al., 2021). Researchers using these devices in sub-Saharan Africa have encountered technical challenges such as faulty synchronization, logistical challenges such as damaged devices, and issues with data completeness (Huhn et al., 2022a).

Unfortunately, essential details of the implementation process of using WATs for research in global south regions such as those in sub-Saharan Africa are not widely documented (Shin et al., 2019; Huhn et al., 2022b). In this perspective, the authors document and reflect on using WATs to conduct psychotherapy research in Kenya. After mentioning some of the challenges faced and strategies to address these challenges, we condense these into lessons learned. Our findings are most relevant to other investigators pursuing global mental health research with WATs.

## Testing a school-based intervention for depression and anxiety

Our multicultural team of researchers has created a character strengths intervention with three modules: gratitude, growth mindset, and values (Osborn et al., 2020). The Shamiri Intervention was delivered to youth with depression or anxiety in Kenyan secondary schools by trained lay providers over 4 weeks in three randomized controlled trials (Osborn et al., 2020, 2021; Venturo-Conerly et al., 2021). A follow-up study of the participants is currently underway to determine the long-term effects of this intervention (Venturo-Conerly et al., 2022).

The Shamiri Intervention took place in secondary schools in Nairobi and Kiambu counties with participants of widely ranging sociodemographic backgrounds, representative of the students attending secondary school in these counties. Some schools taking part in this study were located in informal settlements and enrolled students from the surrounding area, while other schools enrolled students from the entire county. The highest caliber of schools that participants were recruited from enrolled gifted students from the entire country.

As part of the long-term follow-up study, youth were asked to wear WATs for 2 weeks to provide insight into the effects of this treatment on their day-to-day lives. Participants may or may not have had prior experience with WATs, but had limited access to them while at school, as students were not permitted to have electronic devices. Special permission for students to wear WATs in school was obtained. Participants were briefed about the protocol individually or in small groups by a trained research assistant. They were instructed to wear the watches as much as possible and not to share them with others.

The wearables were used to complement self-reported measures with more objective measures of chronic states. The Patient Health Questionnaire, 8-item (PHQ-8) was used for depression symptoms (Kroenke et al., 2009), and the Generalized Anxiety Disorder Screener, 7-item (GAD-7) was used for anxiety symptoms (Spitzer et al., 2006). We used the WATs to capture heart rate variability, distance, and step count. Heart rate variability has been shown to correlate with stress levels and psychological health, while physical activity has been shown to correlate with depressive symptoms (Kim et al., 2018; Moshe et al., 2021).

WATs are an accurate, minimally invasive, relatively affordable, and low-stigma method of gathering ecologically valid, objective data from participants (Evenson et al., 2015; Germini et al., 2022). Although there are many benefits to the use of WATs, they are accompanied by several significant challenges. Through the process of deploying these devices in a pilot study, we learned the following lessons.

## Lessons learned

### *Ensure accurate and available data output*

Securing suitable devices for research in many countries in the global south can pose a significant challenge. In our study conducted in Kenya, the initial study device was selected based on local availability and low cost. However, after data collection began for our pilot study, it became evident that the chosen device, the Xiaomi Mi, did not measure heart rate in frequent, consistent time intervals, which was critical for our study objectives. Moreover, previous research had yielded mixed results regarding the accuracy and validity of the measurements taken with this band (Chow and Yang, 2020; Pino-Ortega et al., 2021; de la Casa Pérez et al., 2022). Thus, following the pilot, we decided to change the study device to the Fitbit Charge.

The Fitbit Charge has a proven track record in large-scale research, especially in measuring heart rate variability (Natarajan et al., 2020), and its accuracy has been demonstrated in various studies (Evenson et al., 2015; Wahl et al., 2017; Germini et al., 2022; Irwin and Gary, 2022). Although the data quality from Fitbits is higher, we found it difficult to extract the data from the devices, specifically, the heart rate variability. To access this data, we required a software developer to build a Fitbit application to request permission from the Fitbit user account of the participant. This gave us access to the data, and then another application was required to compile the extracted data.

To prevent missing or inaccurate data, we recommend researchers validate their chosen devices, ensuring alignment, accuracy, and availability of data. Additionally, before selecting a study device, it is important to ensure the proper technical skills are present to obtain this data. By diligently testing data collection and export procedures, researchers can ensure that the chosen WATs meet their expectations and contribute to robust research.

### *Establish an internal process for international procurement*

Obtaining the necessary study devices in many countries in the global south can present significant challenges, particularly when it comes to their availability in local markets. In our research project, the necessary study devices were not locally available in the required quantity. Thus, we had to navigate the complex processes of sourcing, shipping, and importing, which took 5 months to complete. This delay taught us the necessity of factoring in ample time ahead of planned activities to account for the sourcing, shipping, and customs clearance processes.

In our case working within a new local non-profit, the procurement process for international shipments had not been outlined prior to the WAT purchase. Hence, the WATs used for our study were procured through local vendors who sourced them from several international vendors. This process led to a critical delay in

the receipt of our devices. Thus, we recommend that an organization wishing to purchase a large number of WATs for shipping to a country structurally similar to Kenya establishes an internal international procurement channel. By taking on this process in-house, the organizations will have more control over the procurement and import timeline. This strategy can mitigate the risk of delay and ensure the availability of study devices, facilitating the smooth execution of research projects with wearables.

### Provide onsite support to participants during the data collection and active study period

Deploying WATs for data collection can prove to be an unpredictable endeavor, particularly when working in study sites located in countries in the global south. This unpredictability arises due to technical issues with the devices and participants' varying exposure to technology. For instance, during our pilot study, the devices became unpaired, lost their charge more quickly than anticipated, were reset to factory settings and were often unable to be charged due to lack of access to a power point or electricity.

As the use of WATs in countries in the global south is highly context specific, unanticipated and complex problems may also arise during the active data collection period. In our research endeavors, we encountered several unforeseen issues, highlighting the importance of offering on-site support to participants. Recognizing the significance of this support, we established a dedicated technical and logistical support site, ensuring that participants had access to assistance when encountering challenges with their devices. This approach aligns with the recommendations of those operating WAT research in high-income countries (Smuck et al., 2021).

To facilitate data completeness and prevent disruption, we suggest that during the data collection and active study phase, researchers designate a specific time and location where participants can easily meet a study team member during implementation. This accessibility ensures that participants can seek immediate assistance for any personal, technical or logistical concerns related to the WATs. By providing on-site support, researchers can enhance participant engagement, troubleshoot device-related issues promptly, and promote the completeness of collected data.

### Designate a full-time study team member to operate the wearables

The collection of data using WATs in the global south presents a complex and time-consuming endeavor that heavily depends on the chosen device and context. Adequate allocation of human resources is crucial to their successful deployment. To ensure smooth operations, it is important that there is a study team member who is trained in depth to manage WAT tasks. In the case of our study, we did not anticipate the amount of human support that was required. Due to the intricacies and time-intensive nature of the setup, implementation, and data retrieval, we required a full-time staff member at the peak of activity. When researchers plan to issue WATs to participants as part of the research study, we emphasize the importance of employing a full-time research assistant who possesses expertise in device setup, data extraction, and troubleshooting. This individual plays a critical role to support the completeness and availability of data. Their dedicated focus on managing the intricacies of WAT operations enhances the overall effectiveness and efficiency of data collection efforts.

### Distribute standardized information sheets to participants

Access to electricity and technology can vary drastically for youth in the global south. While it is possible that youth will encounter no challenges adhering to study procedures related to WATs, researchers should thoroughly prepare participants for any potential issues that may arise. The challenges youth in Kenya encountered during our study were with sharing of the devices, charging them and keeping them functional. For instance, many devices became unpaired from the smartphone that they were linked to, rendering them unable to collect data. Regarding the charging of devices, several participants reported that they did not have access to electricity or a charging point. Other participants encountered very rapid battery drainage.

In our study, many youth were excited to wear the WATs. However, as secondary schools in Kenya do not allow electronic devices, wearing the smart watches during school would have made the students stand out. Wearables are popular in Kenya, and many times the youth reported that their friends or family members wanted to borrow the device, which was against the protocol. One participant who was living in a dormitory with other students lost the device. Several youth asked if they could keep the watches following the study period.

While all youth were briefed on study procedures, there was a wide variation in the difficulty youth faced when following them. Researchers using WATs for research with youth in the global south should provide a paper-based information sheet to participants that gives a step-by-step explanation of the procedures, basic troubleshooting instructions, and contact information for the study team in case of challenges. This information sheet should also contain graphics of key messages, *i.e.*, sharing of the watch with others. Provision of this sheet will ensure that the participants have a resource to refer to as they navigate study procedures.

### Conclusion

The use of WATs for treatment research in the global south holds promise as an ecologically valid data collection method. However, researchers must be aware of the challenges they may face when using WATs, such as the time taken to procure devices and manage the implementation of their use, unforeseen technical and logistical failures, as well as data reliability and validity. Strategies to overcome these challenges include to validate data input and output before deployment, establish a trusted and robust international supply channel to procure the devices, train a specific team member to be an expert in WATs, set up onsite technical support for study participants, and provide participants with standardized information sheets.

What we discussed constitutes best practices derived from psychotherapy research in secondary schools with Kenyan youth. These lessons may be most relevant to researchers working with WATs in contexts similar to Kenya. However, certain aspects may apply globally, for example, data extraction. Despite the challenges, data from wearables provides key insight into participant's daily physiological states; an invaluable perspective when considering treatment effects. The unique insights offered by wearable data warrant the difficulty. Thus, we encourage other researchers to consider the benefits of using wearable data alongside careful planning for their use. Ultimately, we believe our findings can contribute to other researchers successfully setting up and undertaking global mental health research with WATs.

**Open peer review.** To view the open peer review materials for this article, please visit http://doi.org/10.1017/gmh.2023.85.

**Acknowledgments.** The authors would like to acknowledge Nina Kahura for her support to implement the use of wearable devices during this study.

**Author contribution.** N.E.J.: conceptualization and original draft manuscript. K.E.V.-C.: supervision and draft manuscript edits. T.R.: supervision and draft manuscript edits.

**Financial support.** WAT devices were funded by Alchemy Pay, a blockchain organization with the mission to bridge fiat and cryptocurrencies. The implementation of study activities was funded by Templeton World Charity Foundation (Grant No. TWCF0633, 2021).

**Competing interest.** K.E.V.-C. is an executive of Shamiri Institute, a 5,013 (c) non-profit organization focused on increasing access to mental health care for youth in sub-Saharan Africa. T.R. is a consultant and member of the Science Board of Shamiri Institute. The remaining author declares no competing interests exist.

**Ethics statement.** This study was approved by Kenyatta University's Ethical Review Committee (PKU/2392/E1528).

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
