## [Reviewer Report]

Cover Letter

Dear Prof. Belkin, Prof. Bass, and Dr. Chibanda, 

My co-authors and I have enclosed for your review a copy of our manuscript entitled, “Using Wearable Activity Trackers in Low-and Middle-Income Countries: Lessons Learned from Adolescent Mental Health Research in Kenya” for your consideration for publication in the journal Global Mental Health. This manuscript has not been published nor submitted for publication elsewhere and was prepared according to the Instructions for Authors provided on the journal website. The use of wearables for this research was funded by Alchemy Pay and a grant from the Templeton World Charity Foundation (Grant Number TWCF0633). 

We believe that our manuscript aligns well with the mission and scope of GMH. In this article, we describe the challenges faced when using wearable activity trackers to determine the long-term health effects of a mental health intervention on youth in Kenya. We then outline strategies that may be useful to other researchers using these devices for research in low-resource settings. 

We believe this manuscript may interest both the readership of GMH focused on designing and testing mental health interventions for youth in low-resource settings and the readership of GMH interested in the integration of wearable activity trackers as a measurement device to collect ecological data in lower- and middle-income countries. All authors have approved the final submitted manuscript and have agreed on submitting it to GMH. 

Below are the authors’ full names and affiliations:

Natalie Johnson

University Hospital Basel

University of Basel

Basel, Switzerland

Shamiri Institute

Nairobi, Kenya

Natalieelaine.johnson@usb.ch

+41 78 243 8166

Katherine Venturo-Conerly

Harvard University

Shamiri Institute, Inc.

Thomas Rusch, PhD

Vienna University of Economics and Business

Best Wishes, 

Natalie Johnson

---

## [Reviewer Report]

This is an excellent article that addresses the challenges (and potential solutions) that many people in LMIC face when trying to use the specified technology in data collection. If we could all collectively find solutions to all these challenges then it will be great for related research in LMIC.

---

## [Reviewer Report]

Thank you for sharing your hard learned lessons. We need more researchers to share this type of experience so that they can, hopefully prevent these challenges.

I am able to understand the target population LMICs, specifically youth in Kenyan secondary schools. I also understand that the WATs were used to complement students self-reported measures with the objective measures attained by the Fitbit Charge.

What I would recommend is for the authors to add more information about the context of the study. For instance, what kind of self-report measures were used? Were the youth trained or experience any difficulties in using or even remembering to wear the WATs? In general, how well did the self-report measures of anxiety and depression agree with the objective measures of Heart Rate and Heart Rate Availability. I recognize that this manuscript is not focusing on results but instead, focuses on lessons learned. One important lesson might also be - for all of the challenges, did the results warrant this approach? Or might you recommend a different approach?

---

## [Reviewer Report]

As the authors rightly state, the use of wearable technologies is on the increase and understanding the potential and limitations of their use in LMICs is essential. I think the manuscript already presents some interesting findings and would be strengthened by some more nuanced discussion. LMICs have quite some national variations as well as diversity. More detailed explanations of country and population level context would be very helpful for readers seeking to apply these lessons.

For example, reading the points about validating data output, procurement and piloting, they ultimately can be reduced to a pilot phase was needed. Pilots are standard in many research projects, particularly those involving technology. Why did the authors not consider that it would be essential to pilot the technology before starting?

I also wonder at the time from procurement, is this really expected to be a standard issue in LMICs? In other LMICs where I’ve worked, devices as common as the Fitbit Charge could easily be bought on Amazon UK or US and delivered in about a month. This makes me curious about the causes of delay? Was it due to numbers needed? Country legal restrictions? Budget constraints? When should researchers expect procurement to be a problem

While the discussion of the technology and challenges related to the use of WATs is important, I missed the human perspective. How did adolescents respond to the technology? What were the reasons they had challenges charging the device -i.e. battery drainage? Why was theft and loss common, context specific or more age group related. Did WATs already exist in the population or was their appearance distinct and possibly something that would make adolescents stand out?

I was also surprised that on-site support is needed for adolescents who tend to be very technology savvy. From my read of Smuck et al.2021., I assume they are dealing with an older population of patients. Did the authors expect this problem with their population, for example are there specifics of this adolescent population that means they might struggle with technology

The authors also mention that collecting data from WATs is time consuming in LMICs. Why is this? I would have thought that the advantage of using WATs would be reducing human effort. What made it difficult in Kenya? Are there possibilities that other LMICs without similar constraints would need less human support?

---

## [Reviewer Report]

Thank you for submitting your manuscript for review. The reviewers have acknowledged the relevance of your work; however, they have requested additional information concerning the context of the study and the setting. They have also posed pertinent questions and provided valuable suggestions that could enhance the manuscript. We kindly request you to address each comment, question, and suggestion in your response for further consideration.

---

## [Reviewer Report]

Dear Profs. Ikenna and Chibanda,

We have enclosed for your review a copy of our revised manuscript entitled, “Using Wearable Activity Trackers in the Global South: Lessons Learned from Adolescent Mental Health Research in Kenya,” MH-23-0138 for your consideration for publication in the journal Global Mental Health. This manuscript has not been published nor submitted for publication elsewhere and was prepared according to the Instructions for Authors provided on the journal website. The use of wearables for this research was funded by Alchemy Pay and a grant from the Templeton World Charity Foundation (Grant Number TWCF0633). 

As stated previously, we believe that our manuscript aligns well with the mission and scope of GMH. In this article, we describe the challenges faced when using wearable activity trackers to determine the long-term health effects of a mental health intervention on youth in Kenya. We then outline strategies that may be useful to other researchers using these devices for research in low-resource settings. 

We believe this manuscript may interest both the readership of GMH focused on designing and testing mental health interventions for youth in low-resource settings and the readership of GMH interested in the integration of wearable activity trackers as a measurement device to collect ecological data in the global south. All authors have approved the final submitted manuscript and have agreed on submitting it to GMH. 

Should you require further changes to the manuscript, please let us know, and we would be glad to make further changes. Below are the authors’ full names and affiliations:

Natalie Johnson

University Hospital Basel

University of Basel

Basel, Switzerland

Shamiri Institute

Nairobi, Kenya

Natalieelaine.johnson@usb.ch

+41 78 243 8166

Katherine Venturo-Conerly

Harvard University

Shamiri Institute, Inc.

Thomas Rusch, PhD

Vienna University of Economics and Business

Best Wishes, 

Natalie Johnson

---

## [Reviewer Report]

Many thanks to the authors for their thoughtful responses to the earlier comments. I think the additional reflections have strengthened this piece and will make it a useful resource for other researchers.

---

## [Reviewer Report]

Thank you for revising the manuscript and responding to the reviewers’ recommendations.

All the reviewers are satisfied with the revisions and have recommended that we accept the manuscript. 

We are happy to accept the manuscript in its present form and look forward to working with you through the publication process